# A Validation Study of the Revised Caregiving Burden Instrument in Korean Family Caregivers of Stroke Survivors

**DOI:** 10.3390/ijerph18062960

**Published:** 2021-03-14

**Authors:** So Sun Kim, Young Sook Roh

**Affiliations:** 1College of Nursing, Yonsei University, Seoul 03722, Korea; soskim396@naver.com; 2Red Cross College of Nursing, Chung-Ang University, Seoul 06974, Korea

**Keywords:** burden, caregivers, reliability, stroke, validity

## Abstract

Background: The purpose of the present study was to examine the internal consistency reliability and construct validity of the Caregiving Burden Instrument in Korean informal caregivers of stroke survivors. Methods: A descriptive survey design was used with a convenience sample of 208 primary caregivers of stroke survivors. Internal consistency reliability was assessed using Cronbach’s alpha coefficients. Construct validity was assessed using exploratory and known-group analysis. Results: Each subscale and the total scale demonstrated satisfactory internal consistency reliability. Exploratory factor analysis identified five factors: family support, patient’s dependency, physical health, financial burden, and psychological health, which together accounted for 62.7% of the variance. Known-group analysis indicated that caregivers with more than one year of experience reported significantly higher mean scores for the total burden score and its five subscales compared to those with less than one year. Conclusions: This 23-item instrument demonstrates good internal consistency reliability and construct validity. The tool can be used to effectively assess burden in caregivers of stroke survivors and the data obtained can form the basis for the development of family interventions.

## 1. Introduction

Stroke is a leading cause of lifelong functional disability that limits the activities of daily living of stroke survivors worldwide [1,2]. Due to the sudden-onset nature of strokes, families are faced with a situation where they have to take care of the patient themselves without being fully prepared [3,4]. As most stroke survivors live in the community and require long-term assistance from family, caring for them induces long-term effects in both stroke survivors and their family caregivers [5,6]. The prevalence of caregiver burden has been found to be 25.0–46.0% [7] and that prevalence remains elevated for an indefinite period following stroke [8,9]. Caregiver burden decreased from baseline to three months, then increased up to nine months [10]. However, this contrasts with results showing no difference in caregiver burden between 2 and 12 months in the Netherlands [9], while caregiver burden in Poland was higher at six months than at five years after stroke [7]. In addition, poor quality of life in stroke caregivers is associated with patient re-hospitalization and increased healthcare costs [10].

Caregiving includes assistance with basic and instrumental activities of daily living, support for addressing medical and treatment requirements, and emotional support and comfort [5]. Caregiver burden has been defined according to various attributes in the extant research. A concept analysis has revealed three important attributes of caregiver burden in the early post-stroke period: (1) objective aspects (caregiving tasks for patient’s assistance, amount of time, costs) and subjective aspects (psychological, social, or emotional impact); (2) time spent caring for stroke survivors; and (3) uncertainty about the future for the stroke survivor and caregiver [11]. Family stroke caregiving has been characterized as physical, psychological, and social suffering resulting from obligation or from a subjective choice, and also as involving a sense of reciprocity [12]. Family caregivers of stroke survivors reported feeling emotionally drained, anxious, uncertain, and apprehensive, practicing unhealthy habits and anticipating challenges during the transition from rehabilitation to home [1]. Caregiving was interpreted as suffering, an obligation, a personal choice, a meaningful opportunity, and a natural part of living in Chinese family caregivers of stroke survivors [13]. While most studies emphasized the negative attributes of caregiving, some studies also reported positive attributes, such as a sense of reciprocity [12], meaningful opportunity [13], and role gain [14].

Several tools have been used to measure burden in caregivers of stroke survivors [11]. The 22-item Caregiver Burden Scale has five subscales: general strain, isolation, disappointment, emotional involvement, and environment. Each subscale demonstrated satisfactory reliability, except for environment, among 150 Swedish family caregivers of patients with dementia or stroke [15]. The 27-item Sense of Competence Questionnaire comprises three subscales with good reliability, as demonstrated among 166 Dutch partners of stroke patients: satisfaction with care recipient, satisfaction with caregiving performance, and impact on personal life. The tool was shown to explain 42.0% of the total variance of caregiver burden [16]. The Bakas Caregiving Outcomes Scale, revised from 10 to 15 items, is a one-factor scale tested with a sample of 147 American family caregivers of stroke survivors. The scale showed satisfactory internal consistency, test–retest reliability, and construct validity, with 42.8% of the variance accounted for by the single factor [17]. Other tools are designed to measure the burden among caregivers of older adults with cognitive impairments (Zarit Burden Interview) or persons with dementia (Relatives Stress Scale), physical impairments (Caregiver Strain Index), or cancer (Caregiver Reaction Assessment), which raises the issue of validity when measuring the burden of caregivers of stroke survivors. Unlike Alzheimer’s dementia, which worsens over time, stroke survivors may experience periods of stability [15]. Therefore, careful consideration should be given to the selection of measurement tools for caregivers of stroke survivors.

As there is a prehospital delay depending on the socioeconomic status and ethnic group of stroke patients [18], it is necessary to fully consider the contexts of the patient and their family to understand the caregiving experience. Caregiving experiences vary depending on the cultural background of the caregivers. In an earlier study, caregiving was perceived to stem from either a sense of familial duty, associated with cultural background, or an individual choice made by members of diverse ethnic groups [19]. For instance, non-African American caregivers have reported more negative caregiving outcomes than African American caregivers [20]. Cultural factors, such as Confucianism and filial piety, can influence Chinese stroke caregivers’ role perception [21,22]. Koreans have a similar familial obligation, owing to Confucianism and filial piety, which differs from Western cultures. Generalizability of earlier instruments, developed with predominantly Western values in mind, is therefore limited in the Korean caring context.

Caregiver burden has largely been investigated using quantitative measures, an approach that can sometimes fail to capture contextual and cultural features that are relevant to caregiving outcomes [23]. Therefore, there is a need for a concise tool that can isolate the influence of culture on caregiving experiences. The Caregiving Burden Instrument is a known tool that can be used to rate the consequences of caregiving, reflecting values and norms of caregiving experiences among Korean caregivers of stroke survivors. It was developed on the basis of a literature review and in-depth interviews with Korean caregivers of stroke survivors who felt strong familial obligation owing to Confucianism and filial piety [24]. A reliable and valid instrument is needed for accurate assessment of caregiver burden in order to prioritize and address the needs of stroke caregivers; however, there is little empirical validation on measurement structures of caregiver burden. The purpose of the present study was to examine the internal consistency, reliability, and construct validity of the revised Caregiving Burden Instrument in Korean family caregivers of stroke survivors using an exploratory factor analysis and known-group analysis based on a one-year caregiving period. The caregiving period was selected based on a study noting that caregiver burden should be measured repeatedly until 12 months after a stroke [2].

## 2. Materials and Methods

### 2.1. Design 

A descriptive survey design was used.

### 2.2. Participants

A factor analysis requires a minimum of five participants per initial item [25]. Thus, with 35 items, a minimum of 175 participants were needed for this study. A convenience sample of 246 Korean family caregivers was recruited from outpatient departments at two hospitals located in Seoul. Inclusion criteria were that participants should be family caregivers over 20 years of age taking care of stroke patients living in the home. A total of 208 questionnaires were analyzed, with 38 incomplete questionnaires excluded.

### 2.3. Measurements

#### 2.3.1. Demographics of Caregivers and Stroke Survivors

The questionnaire covered the demographic information of the family caregivers (age, gender, marital status, education, religion, relationship with care recipient, caregiving duration) and stroke survivors (age, gender, marital status).

#### 2.3.2. Caregiver Burden

Caregiver burden was measured using the Korean version of the Caregiving Burden Instrument [24]. This previously developed instrument rates the consequences of caregiving, such as social activity, family support, patient’s future, caregiver’s future, financial status, patient’s dependency, and physical health, in caregivers of stroke survivors [24]. The conceptual framework for caregiver burden was developed based on a literature review [14,15,16] and includes five core dimensions of caregiving burden, each with two attributes: patient’s dependency, physical health, psychological health, social health, and financial status (Figure 1). The scale development process has been reported in a published document [24]. Exploratory factor analysis for the preliminary 35-item scale identified seven factors accounting for 63.9% of the total variance: social activity, family support, patient’s future, caregiver’s future, financial status, patient’s dependency, and physical health. The Cronbach’s alphas of these seven factors ranged from 0.55 to 0.92. The reliability of the total scale was previously demonstrated by a Cronbach’s alpha of 0.93. The preliminary 35-item scale available for the current study asked participants to rate their responses on a five-point Likert-type scale ranging from 1 (strongly disagree) to 5 (strongly agree). A higher score indicates higher caregiver burden [24].

### 2.4. Ethical Considerations

This research was approved by the Institutional Review Board of the Yonsei University College of Nursing (IRB 2015-0001-2). Informed consent was obtained from each family caregiver who volunteered. Respondent anonymity was maintained throughout the data collection and analysis.

### 2.5. Data Collection

The author contacted the nursing departments in two hospitals in Seoul to request their participation in the research. A self-administered questionnaire was distributed to all eligible family caregivers of stroke survivors by a well-trained research assistant. The data were collected from 1 April to 31 December 2017.

### 2.6. Data Analysis

Data were analyzed using IBM SPSS Statistics for Windows, Version 23.0 (IBM Corp., Armonk, NY, USA). The item means and standard deviations, inter-item correlation matrix, and item-total correlations were computed for item analysis. Cronbach’s alpha coefficients were calculated to assess internal consistency reliability. The Kaiser–Meyer–Olkin (KMO) measure of sampling adequacy and Bartlett’s test of sphericity were also performed. Exploratory factor analysis was carried out using principal component analysis with varimax rotation to assess the instrument’s construct validity. Known-group analysis comparing more than one year and less than one year of caregiving experience was performed using two-sample *t*-tests.

## 3. Results

### 3.1. General Characteristics

Of the 246 questionnaires, a total of 208 questionnaires were analyzed, excluding 38 incomplete questionnaires. The mean age of the stroke survivors was 65.85 years (SD = 15.86); the largest category was those in their 60s and 70s (n = 57, 27.4%). Of the 208 patients, a total of 134 (64.4%) were men and 146 (70.2%) were married. Of the 208 family caregivers, the mean age was 57.77 years (SD = 13.20); the largest category was those in their 60s (n = 64, 30.8%). A total of 172 (82.7%) were women, 95 (42.8%) were college graduates, 176 (84.6%) were married, and 93 (44.7%) were spouses of stroke survivors. One hundred forty-three participants (68.8%) had a monthly income of less than $4500. The mean caregiving duration was 71.55 months (SD = 79.67) (Table 1).

### 3.2. Psychometric Testing of the Scale

#### 3.2.1. Item Analysis

The item means and standard deviations, inter-item correlation matrix, and item-total correlations were computed and examined. The item-to-total scale correlations of the preliminary 35-item scale ranged from 0.42 to 0.79. Nine items were eliminated because of redundancy or lack of homogeneity with the construct; corrected item-total correlations were taken to be acceptable if they ranged from 0.30 to 0.70 [26]. The remaining 26 items had corrected item-total correlation coefficients between 0.41 and 0.68.

#### 3.2.2. Examination of Factor Independence and Reliability

On a five-point scale, the mean caregiver burden score was 3.51 ± 0.64. Using the results of a previous study [27], the top 25th percentile, 3.89, was defined as the cut-off score. Of the total of 208 family caregivers, 53 (25.5%) had scores above the cut-off score of 3.89. The reliability estimates of the five factors ranged from 0.68 to 0.89. The Caregiver Burden Instrument demonstrated high internal consistency, with an alpha value of 0.93. The correlations between the five factors ranged from 0.39 to 0.64.

#### 3.2.3. Exploratory Factor Analysis

Prior to performing the analysis, the suitability of the data for factor analysis was assessed. Bartlett’s test of sphericity was significant (χ^2^ = 2649.68, *p* < 0.001), and the Kaiser–Meyer–Olkin (KMO) measure of sampling adequacy was appropriate at 0.92. Three items were deleted—two that were cross-loaded >0.40 on both factors (“The patient cannot do anything unless I help him” and “I am worried about what will happen to the patient’s prognosis”) and one that had factor loading of less than 0.40 (“I want to live the life I hoped I would have”). Factors were extracted based on a minimum eigenvalue of 1 as a cut-off value [26]. Factors were interpreted and labeled based on the five main themes of family caregiver burden: “family support”, “patient’s dependency”, ”physical health”, “financial burden”, and “psychological health”. These five factors explained 62.7% of the total variance (Table 2).

#### 3.2.4. Construct Validity: Known-Group Analysis

On a five-point scale, family caregivers with more than one year of experience reported significantly higher means than those with less than one year experience for total burden (3.60 ± 0.62 vs. 3.22 ± 0.59, *t* = −3.94, *p* < 0.001), family support (2.70 ± 0.85 vs. 2.09 ± 0.80, *t* = −4.58, *p* < 0.001), patient’s dependency (3.90 ± 0.74 vs. 3.52 ± 0.71, *t* = −3.30, *p* = 0.001), physical health (4.00 ± 0.72 vs. 3.75 ± 0.68, t = −2.23, p = 0.027), financial burden (3.65 ± 0.87 vs. 3.35 ± 0.83, *t* = −2.70, *p* = 0.007), and psychological health (3.76 ± 0.79 vs. 3.42 ± 0.78, *t* = −2.18, *p* = 0.030). The caregiving period was selected based on a study that suggested that caregiver burden should be measured repeatedly until 12 months after a stroke [2].

## 4. Discussion

Our findings demonstrate the internal consistency, reliability and construct validity of the revised Caregiving Burden Instrument. The development of a sound instrument is useful for the appraisal of family caregiver burden and necessary for helping caregivers cope better with the care of stroke survivors. Exploratory factor analysis identified five factors termed “family support”, “patient’s dependency”, “physical health”, “financial burden”, and “psychological health”. These five factors accounted for 62.7% of the variance, meeting the threshold of 60.0% or higher of variance explained [28]. In contrast, the Sense of Competence Questionnaire explained 42.0% of the total variance of caregiver burden [16], and the 15-item Bakas Caregiving Outcomes Scale explained 42.8% of the variance [17].

The factors identified in our study were similar to those identified in previous studies on caregiver burden, such as objective and subjective aspects, time spent caring for the stroke survivors, and uncertainty about the future of the stroke survivor and the caregiver [11]; or physical, psychological, and social suffering, with caregiving conceived as an obligation or a subjective choice [12]. These factors were also similar to some factors that have been defined as multidimensional variables [15,16,29] or attributes [1]. In contrast to a previous instrument, which did not cover physical burden [29], our instrument included a “physical health” factor, as the Korean family caregivers in our study felt a physical burden due to caring for stroke survivors with their decreased mobility and cognitive impairment. One study suggested positive indicators, such as role gain, in caregiving [14], whereas mainly negative indicators were identified in this study. On the basis of the present findings, caregiver burden for stroke caregivers can be defined as the physical, psychological, social, financial, and time-related burdens that arise from direct care of dependent stroke survivors who require support in daily life because of physical and/or cognitive disabilities.

The five-factor Caregiving Burden Instrument showed satisfactory internal consistency, with a total Cronbach’s alpha of 0.93, above the recognized threshold of 0.70 [30]. In contrast, Cronbach’s alpha ranged from 0.53 to 0.87 for the Caregiver Burden Scale [15] and was 0.83 for the Sense of Competence Questionnaire [16]. Therefore, the revised Caregiving Burden Instrument in our study is a reliable tool to measure burden among caregivers of stroke survivors.

The initial Caregiving Burden Instrument was a 35-item tool that included questions on the stroke caregiver’s social activity, family support, the patient’s future, the caregiver’s future, financial status, the patient’s dependency, and physical health [24], which were systematically refined to 23 items in the present study. The Burden Interview (BI), which has been extensively used in studies investigating caregivers of people with dementia, was also refined, from a 29-item questionnaire to a 12-item screening tool [31]. To date, various measures have been used in caregiver burden studies, which may raise the issue of validity. The 23-item Caregiving Burden Instrument as revised in our study reflects the cultural perspectives and characteristics of stroke survivors, as well as their role as a whole.

A limitation of the most popular burden instruments is that they were developed for caregivers of people with dementia or other physical morbidities, leading to the need for caution when applying them to caregivers of stroke survivors [24]. The nature of caregiver burden for strokes may be different from that for these other ailments because, unlike a progressive disease, such as dementia or other chronic diseases, stroke is a sudden-onset neurological disease that can demand family caregiver effort without any preparation [3,4]. Caregiver burden with stroke survivors is often related to their physical functioning [10] and is also associated with the amount of time caregivers spend caregiving [7,32]. In addition, there was relatively low consistency in attributes among tools for measuring burden among caregivers of stroke survivors [11].

“Family support” was found to be the factor that explained the most variance in the instrument. Similarly, the Caregiver Burden Scale included “relational burden” [29]. In Korea, family-centered collectivism is recognized as a basic principle of society, originating with the influence of the Confucian value system, and the family-based support system significantly affects the burden of the caregiver [24]. A similar phenomenon has been found in some other Asian countries [21,22]. Thus, our instrument reflects and upholds the notion that caregiver burden measures should be more context-sensitive, considering local cultural notions of familial or filial obligation, role overload, and role conflict [23].

Family caregivers with more than one year of experience, compared to those with one year or less, reported significantly higher mean scores for total burden score and its five subscales. This result supports the interpretation that the burden of care is positively related to the length of caregiving [32]. Caregiver burden decreased from baseline to three months, then increased up to nine months [10]. However, this contrasts with results showing no difference in caregiver burden between 2 and 12 months in the Netherlands [9], while caregiver burden in Poland was higher at six months than at five years after stroke [7]. Among the five factors, family support had a higher level of burden in family caregivers with care periods of 12 months or more. This may have been due to the strong recognition of family support in caregiving among Koreans. As the stroke survivor’s illness becomes prolonged, the degree of concern and support by other family members decreases compared to the early stage of the onset. In this study, the burden of long-term family caregivers was high; therefore, it is necessary to apply customized interventions based on their actual demonstrated needs.

The strength and the major contribution of this study is its establishment of satisfactory internal consistency reliability and construct validity in the revised Caregiving Burden Instrument, enabling accurate assessment of family caregiver burden among those caring for stroke survivors, and reflecting the limitations of existing burden measurements and the need for culturally nuanced perspectives. The study limitations are as follows. First, as this research was conducted among Korean family caregivers, it offers limited scope for generalization in other cultures. Further refinement of the instrument in other cultures could benefit the caregiving research area. Second, in this study, only the internal consistency reliability of the instrument was verified; another verification of test–retest reliability is needed. Third, as predictive validity and convergent validity were not verified for this tool, further psychometric testing is needed in clinical settings. Fourth, the psychological or mental health status of caregivers was not assessed in this study. Given the fact that long-term disability is associated with stroke survivors and that caregivers have substantial responsibilities, this may also be an important factor to consider. Finally, because this study did not investigate the disease characteristics of stroke patients, such as whether a stroke patient had a first-onset stroke or a secondary stroke, the study results should be interpreted with caution.

## 5. Conclusions

Family caregivers are exposed to various crises for which they are unprepared due to the sudden onset of strokes. Stroke survivors require ongoing family support due to sudden dysfunction, and family caregivers’ caregiving contributes to the success or failure of rehabilitation of stroke survivors. Nurses play important roles as information providers, educators, or coordinators for family caregivers of stroke survivors by assisting or supporting them across the care spectrum. Thus, it is important for nurses to identify the needs of families facing crisis and apply effective intervention programs. The results of this study showed that 25.5% of family caregivers of stroke survivors indicated a high level of burden and burden was higher after 12 months of caregiving. Therefore, it is necessary for nurses to identify high-risk family caregivers who experience a high caregiving burden through the use of this tool early in the caregiving trajectory, apply an intervention program based on their needs, and follow up the outcomes of intervention. Additionally, multifaceted family educational programs or support interventions are needed to reduce the burden of caring for the family caregiver and strengthen their coping strategies. Further research is needed to evaluate the outcomes of family caregiver interventions based on the priority areas of family caregiver burden management using this instrument. Furthermore, it is necessary to verify the construct validity using confirmatory factor analysis in the caregivers of patients with strokes because the construct validity of the tool was verified through exploratory factor analysis in the present study.

## Figures and Tables

**Figure 1 ijerph-18-02960-f001:**
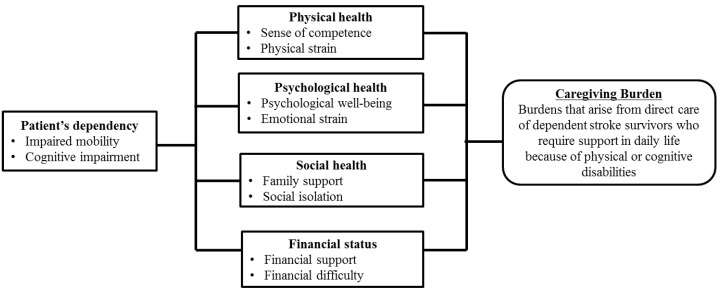
Conceptual framework for caregiving burden.

**Table 1 ijerph-18-02960-t001:** General characteristics of participants (N = 208).

Characteristics		Category	n (%)/Mean ± SD
Stroke survivors	Age (years)		65.85 ± 15.86
		≤30	8(3.8)
		31–40	8(3.8)
		41–50	16(7.7)
		51–60	31(14.9)
		61–70	57(27.4)
		71–80	57(27.4)
		≥81	31(14.9)
	Gender (men)		134(64.4)
	Marital status	Single	29(13.9)
		Married	146(70.2)
		Others	33(15.9)
Caregivers	Age (years)		57.77 ± 13.20
		≤30	7(3.4)
		31–40	18(8.7)
		41–50	32(15.4)
		51–60	56(26.9)
		61–70	64(30.8)
		71–80	27(13.0)
		≥81	4(1.9)
	Gender (women)		172(82.7)
	Marital status	Single	25(12.0)
		Married	176(84.6)
		Others	7(3.4)
	Education	≤High school	102(49.0)
		College	95(45.7)
		Graduate school	11(5.3)
	Religion	None	42(20.2)
		Yes	166(79.8)
	Relationship	Spouse	93(44.7)
		Son	25(12.0)
		Daughter	41(19.7)
		Daughter-in-law	2(1.0)
		Granddaughter	5(2.4)
		Others	42(20.2)
	Monthly income ($)	<2500	80(38.5)
		2500–4499	63(30.3)
		4500–5999	34(16.3)
		≥6000	31(14.9)
	Caregiving duration (months)		71.55 ± 79.67
		≤12	62(29.8)
		13–24	27(13.0)
		25–36	13(6.3)
		37–48	17(8.2)
		≥49	89(42.8)

**Table 2 ijerph-18-02960-t002:** Exploratory factor analysis and the reliability of the subscales (N = 208).

Item (Number of Items)	Factor 1	Factor 2	Factor 3	Factor 4	Factor 5
Factor 1. Family support (7)					
Q1. I feel like a victim of my family.	**0.81**	0.11	0.12	0.11	0.13
Q2. My family does not know about my effort or hard work.	**0.78**	0.13	0.16	0.09	−0.03
Q3. Families are not interested in caring for patients.	**0.78**	0.23	−0.05	0.01	−0.07
Q4. I am frustrated with families who do not care for patients.	**0.77**	0.16	0.06	0.07	0.09
Q5. I get angry when I see a patient.	**0.74**	0.05	0.11	0.19	0.34
Q6. Because of the patient, my relationship with my family is worse than before.	**0.59**	0.23	0.25	0.33	0.08
Q7. I get angry easily because I cannot tolerate things as before.	**0.56**	0.21	0.17	0.35	0.24
Factor 2. Patient’s dependency (5)					
Q8. The patient wants me to stay with him (her) all the time.	0.21	**0.74**	0.11	0.14	0.13
Q9. The patient wants me to do what they can do themselves.	0.25	**0.70**	0.06	0.11	0.12
Q10. Even if I go out, my mind is with the patient.	0.09	**0.61**	0.36	0.24	0.26
Q11. I’ve had fewer meetings recently with people than ever before.	0.17	**0.57**	0.12	0.29	0.31
Q12. I always live in tension because of the patient.	0.18	**0.55**	0.36	0.13	0.40
Factor 3. Physical health (4)					
Q13. I am sorry I cannot do for the patient as much as I want to because I am tired.	0.09	0.07	**0.77**	0.11	0.04
Q14. I am concerned about the future of the patient.	0.07	0.29	**0.69**	0.14	0.27
Q15. I am responsible for my patient beyond my ability.	0.21	0.03	**0.65**	0.21	0.17
Q16. I cannot sleep well enough.	0.12	0.32	**0.55**	0.19	0.02
Factor 4. Financial burden (4)					
Q17. I am worried about the high cost of patient care.	0.02	0.05	0.37	**0.76**	−0.02
Q18. The cost of the patient makes it difficult for the family.	0.21	0.35	0.16	**0.75**	0.17
Q19. I cannot afford to spend money like I used to.	0.28	0.25	0.12	**0.73**	0.21
Q20. I have to do other things in the house while I look after the patient.	0.24	0.19	0.06	**0.43**	0.37
Factor 5. Psychological health (3)					
Q21. I am always worried about the patient.	−0.06	0.16	0.31	0.05	**0.78**
Q22. It is painful to see the patient.	0.39	0.18	0.10	0.32	**0.57**
Q23. When I think about the patient, I get depressed.	0.34	0.33	0.05	0.25	**0.51**
Mean ± SD	2.55 ± 0.88	3.75 ± 0.77	3.98 ± 0.67	3.58 ± 0.87	3.67 ± 0.80
Cronbach’s alpha	0.89	0.85	0.79	0.80	0.68
% of variance	17.7	13.5	11.9	10.6	9.0
Cumulative %	17.7	31.2	43.1	53.7	62.7

Note: Bolded loadings highlight item allocation for each factor.

## Data Availability

Data are available upon request.

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
