# Peer review of "A Validation Study of the Revised Caregiving Burden Instrument in Korean Family Caregivers of Stroke Survivors"

_ijerph, 2021, doi:10.3390/ijerph18062960_

Round 1
Reviewer 1 Report
The purpose of the study descripted in this manuscript was to examine the psychometric properties of the Caregiving Burden Instrument in Korean caregivers of stroke survivors. The focus of this study on the importance of cultural differences in caregiving is a substantial strength.
One thing that is hard to follow throughout the manuscript is whether the factor structure was established via EFA before this study or if this was the first EFA of this measure ever. There are EFA results in the “measures” section of the methods, but then these seem to match the EFA results presented in the results, so it isn’t clear. Also, if EFA has been done before, then why do an EFA (vs just a CFA) in this sample? If this is the first EFA, then it is not appropriate to run a CFA in the same sample (in general, it is not appropriate to do EFA and CFA in the same sample, unless the sample is split in two).
Abstract:
- Indicate known-group analysis groups
- What are the implications of the poor goodness-of-fit and normed fit indices?
Introduction:
- Line 37: including an example of what an objective and subjective aspect of caregiver burden would be would help make this clearer.
- Lines 38-40: This sentence is confusing. As written, it suggests that caregiving=suffering.
- The second paragraph covers the negative aspects of caregiving, but what about the positive aspects of caregiving? Not all caregivers experience burden or perceive caregiving as suffering.
- The paragraph on caregiver burden measures does not include the Zarit Burden Interview, which is a widely used and validated measure – authors may want to consider adding it to this paragraph (it is brought up in the Discussion, however).
- Justification for the “known-groups” should be stronger in the introduction. Why is it that you believed the scale should differentiate those with <1 vs >1 year of caregiving experience? There should be some literature supporting this hypothesis and a clearly stated hypothesis related to the known groups validity.
Methods:
- What language was the Caregiver Burden Instrument collected in for this study?
- Line 136: known-group analysis for? (state <1 vs >1 year experience caregiving). Did you account for those who may have been caregivers before this stroke (were these patients all 1st time stroke patients?)?
- Was CFA performed in the same sample? Or was the sample split?
- It was difficult to follow the results because the methods do not clearly outline the analysis steps. Did the item analysis and reliabilities estimates come before the EFA? A step by step outline of the analysis in the data analysis section is needed, beyond just stating that certain analyses were performed.
Results:
- Centering the far left column of the table makes it difficult to follow. Consider left justifying and indented the categories under each subheading (e.g., single, married, etc. indented under Marital status).
- So it had 7 factors in a previous population and only 5 in this one? How were number of factors determined?
- CFA results: you cannot confirm the factor structure in the same dataset that you established it in. Unless I misread and this was a novel sample, then the CFA should be removed altogether and recommended as a future direction.
Discussion
- The language in the discussion is awkward, making it difficult to understanding sometimes what the authors are trying to say. Close review for language clarity and flow would be helpful. Some points, like that made in lines 251-254 about stroke being a sudden onset condition, are important and should be added to the introduction as justification for looking at a caregiver burden measure specifically in stroke.
Line 269: by “most important” factor, do you mean the one that explained the most variance?
Author Response
Title: A validation study of the revised caregiving burden instrument in Korean family caregivers of stroke survivors
My coauthor and I thank the reviewers for reading and commenting on our manuscript. We now humbly request that they reconsider the attached manuscript, entitled “A validation study of the revised caregiving burden instrument in Korean family caregivers of stroke survivors,” for publication as an original research article in the International Journal of Environmental Research and Public Health. Based on the comments from the reviewers, we have reorganized and revised the manuscript. Changes have been highlighted within the document using colored texts.

Reviewer 2 Report
The authors' findings about “A validation study of the revised caregiving burden instrument in
Korean family caregivers of stroke survivors” are very interesting. The paper is clearly and concisely written. However, there are some concerns regarding preparation of this manuscript as follows:
Minor revisions:
- Author must provide list of Abbreviations that all are used in manuscript for readers. It is difficult to follow short forms in manuscript if full list of Abbreviations is not available.
- In methods section 2.2 Participants, Author state that on page 2, line no 94 about 246 Korean family caregivers and on page 3 line no 96 mentioned about 208 questionnaires analyzed.
When you check Table 1 on page 4, It mentioned N=208
So, participants are less compared to mentioned from methods or this need to clarify clearly how many participants as stroke survivors and caregivers.
- Also, in table 1 and in results section 3.1 stated that A total 134 were men and 146 married.
So, how could married no is bigger than total no of 134 men of stroke survivors?
Similarly, A total 172 were women and 176 were married? This did not make sense on based on questionnaires data.
Author Response

(The authors gave the same response as above.)

Reviewer 3 Report
This is an interesting study on the consistency of Caregiving Burden Instrument in Korean informal caregivers of stroke survivors. The study has merits given its focus on a rather understudied aspect of stroke care - the caregiver. I have few suggestions for authors to consider; 1. On Page 2, second paragraph, authors should also include a sentence or two on the role socioeconomic factors play in influencing various aspects of stroke care (see https://pubmed.ncbi.nlm.nih.gov/31097575/) 2. It is not clear in the methods whether the stakeholders or caregivers were consulted prior to the application of the questionaire. This is poignant to address linguistic appropriateness to the population. 3. Caregiver burden may vary depending on the socioeconomic status. It is not clear if the data on the income status or other such variables are available on this study population. 4. Psychlogical or mental health status of caregivers was not assessed in this study? Can the authors comment if this may also be a factor? Given the long-term disability associated with stroke survivors and the responsibilities on the caregivers - this may also be a an important factor to consider. This should be added as a limitation to the study, if applicable. 5. Conclusion Section: Indeed, Identifying high-risk caregivers is key to develop tailored interventions. Could the authors comment on the broader implications of these findings from the context of medical or non-medical interventions that may be provided to those at high-risk.Author Response
Title: A validation study of the revised caregiving burden instrument in Korean family caregivers of stroke survivors
My coauthor and I thank the reviewers for reading and commenting on our manuscript. We now humbly request that they reconsider the attached manuscript, entitled “A validation study of the revised caregiving burden instrument in Korean family caregivers of stroke survivors,” for publication as an original research article in the International Journal of Environmental Research and Public Health. Based on the comments from the reviewers, we have reorganized and revised the manuscript. Changes have been highlighted within the document using colored texts.

Round 2
Reviewer 1 Report
Thank you to the authors for their thoughtful revision and responses. The manuscript provides an important contribution to field supporting caregivers of individuals with stroke.
Reviewer 3 Report
The authors have adequately addressed all the concerns/comments. I recommend the article be accepted in its current form.